# NN-RNALoc: Neural network-based model for prediction of mRNA sub-cellular localization using distance-based sub-sequence profiles

**Negin Sadat Babaiha**[1,2]**, Rosa Aghdam**[3,4]*****, Shokoofeh Ghiam**[3]**, Changiz Eslahchi** ![ORCID][1,3]*****

**1** Department of Computer and Data Sciences, Faculty of Mathematical Sciences, Shahid Beheshti University, Tehran, Iran, **2** Bonn-Aachen International Center for Information Technology (B-IT), University of Bonn, Bonn, Germany, **3** School of Biological Sciences, Institute for Research in Fundamental Sciences (IPM), Tehran, Iran, **4** Wisconsin Institute for Discovery, University of Wisconsin-Madison, Madison, WI, United States of America

* ch-eslahchi@sbu.ac.ir (CE); rosaaghdam@gmail.com (RA)

## Abstract

The localization of messenger RNAs (mRNAs) is a frequently observed phenomenon and a crucial aspect of gene expression regulation. It is also a mechanism for targeting proteins to a specific cellular region. Moreover, prior research and studies have shown the significance of intracellular RNA positioning during embryonic and neural dendrite formation. Incorrect RNA localization, which can be caused by a variety of factors, such as mutations in trans-regulatory elements, has been linked to the development of certain neuromuscular diseases and cancer. In this study, we introduced NN-RNALoc, a neural network-based method for predicting the cellular location of mRNA using novel features extracted from mRNA sequence data and protein interaction patterns. In fact, we developed a distance-based sub-sequence profile for RNA sequence representation that is more memory and time-efficient than well-known k-mer sequence representation. Combining protein-protein interaction data, which is essential for numerous biological processes, with our novel distance-based subsequence profiles of mRNA sequences produces more accurate features. On two benchmark datasets, CeFra-Seq and RNALocate, the performance of NN-RNALoc is compared to powerful predictive models proposed in previous works (mRNALoc, RNATracker, mLoc-mRNA, DM3Loc, iLoc-mRNA, and EL-RMLocNet), and a ground neural (DNN5-mer) network. Compared to the previous methods, NN-RNALoc significantly reduces computation time and also outperforms them in terms of accuracy. This study's source code and datasets are freely accessible at https://github.com/NeginBabaiha/NN-RNALoc.

## Introduction

Numerous studies have implicated the intracellular localization of RNA as a cellular polarization mechanism, as it plays a crucial role in gene expression regulation [1]. Additionally, messenger RNAs (mRNAs) localization may be preferable to protein localization because a single mRNA molecule can serve as a template for multiple proteins. Therefore, the prediction of mRNA localization rather than protein localization is more efficient and saves time. Recent research suggests, however, that the localization of mRNAs to specific sub-cellular

**Data Availability Statement:** All data are available from the mentioned database in the paper and are also freely available in: https://github.com/NeginBabaiha/NN-RNALoc.

**Funding:** The author(s) received no specific funding for this work.

**Competing interests:** The authors have declared that no competing interests exist.

compartments may be more common than previously thought, and that signals in the sequences of both types of molecules can play a role in their transport to specific cellular locations [2–5]. Notable is the association between incorrect RNA localization within the cell and neuromuscular disorders and cancer. Previously, oligonucleotides were introduced as a new type of drug that targets RNAs rather than disease-causing proteins [6–8]. On the other hand, mRNA localization has been studied for many years, and there are two well-known experimental datasets in this regard: cell fractionation with RNA-sequencing (CeFra-Seq) and APEX-RIP [9, 10]. CeFra-Seq is a method for mapping the abundance of transcripts in the Nucleus, Cytoplasm, Membrane, and Insoluble fractions of cells. APEX-RIP is a technique for mapping Nuclear, Cytoplasmic, Endoplasmic Reticulum (ER), and Mitochondrial transcriptomes. Additionally, RNALocate is a well-known RNA localization dataset. RNALocate is a web-accessible dataset containing information for over 190,00 RNA-associated sub-cellular localization entries supported by experimental and predicted evidence [11]. It involves over 105,000 RNAs in 44 subcellular locations in 65 species, including Homo sapiens, Mus musculus, and Saccharomyces cerevisiae. The gap between existing mRNAs and those whose location is known is increasing the need for computational predictors despite experimental efforts. In recent years, computational predictors have emerged that rely heavily on machine learning techniques [12–14]. RNATracker [9] was the first mRNA localization prediction model to be developed in 2019. RNATracker predicts the location of mRNAs in CeFra-Seq and APEX-RIP datasets using convolutional neural network (CNN) and long short-term Memory (LSTM). In another recent study, a predictive model named RNA-GPS was introduced to predict the localization of the transcripts in the APEX-RIP dataset only [14]. RNA-GPS computes k-mer frequencies for k ranging from 3 to 5 for each transcript and assigns probabilities to mRNA-location using a random forest model. In 2020, mRNALoc was designed to predict mRNA sub-cellular localization by extracting k-mer profiles from mRNA sequences and applying a support vector machine (SVM) and was trained on the RNALocate dataset [12]. Zhang et al., developed a computational method, iLoc-mRNA, which was trained on the RNALocate dataset and applied a SVM model for multiclass classification [13]. It is also noteworthy to mention that in iLoc-mRNA, predictions were made for one of the following locations: Cytosol/Cytoplasm, ribosome, Endoplasmic Reticulum, and nucleus/exosome dendrite/mitochondrion. It is understood that combining nucleus, exosome, dendrite, and Mitochondria as a single location is not appropriate as these are diverse locations which should not be merged into a single sub-cellular class [12]. Meher et al. presented "mLoc-mRNA" to forecast nine distinct sub-cellular localizations for mRNAs. They used k-mers of sizes 1–6 to transform each mRNA sequence into a numerical feature vector. They applied the Elastic Net statistical model to extract the best features from the k-mer features. The sub-cellular localization of mRNAs was then predicted using a Random Forest classifier [15]. In 2021, a multi-label mRNA sub-cellular localization predictor named "DM3Loc" was also proposed using Deep Learning, which predicts the 6 distinct locations of mRNAs in Homo sapiens. They prepared data as the input for CNN using mRNA sequences as the raw data and a novel multi-head self-attention mechanism capable of producing sequence motifs [16]. The deep learning model "EL-RMLocNet", which predicts the subcellular localization of four different RNA classes (mRNA, miRNA, lncRNA, and snoRNA) in Homo sapiens and Mus musculus species, was developed in [17]. To identify the most informative features from raw RNA sequences, they used the LSTM network, which captured the short and long range relations of nucleotide k-mers. In this study, we focus on the CeFra-Seq and RNALocate datasets, as well as powerful predictive models including mRNALoc, RNATracker, mLoc-mRNA, DM3Loc, iLoc-mRNA, and EL-RMLocNet as benchmarks. The rationale for selecting these methods and datasets is that the APEX-RIP dataset is noisy [12]. We also study a ground neural network (DNN-5mer) that only has two hidden

layers and extracts k-mer features from sequences. We presented a novel representation of mRNA sequences based on subsequences of distance k, and we argued that combining this encoding with conventional k-mer frequency profiles can potentially yield more sequence-based information from mRNAs. Using a protein-protein interaction (PPI) network, we developed a neural network-based model that we call NN-RNALoc. Indeed, we utilized the fact that proteins with similar PPI patterns tend to be primarily located in the same sub-cellular location [18, 19] by incorporating this widely-used data into the predictive model. Additionally, it is essential to note that "Chou's 5-step rule" [20] can be used to develop a more practical predictor for a biological system. The Chou's 5-step rules have the following notable benefits: clear in logic development, completely transparent in operation, easily repeatable by other researchers, with a high potential for stimulating other sequence-analysis methods, and very user-friendly for the vast majority of experimental scientists [21, 22]. Therefore, in this study, we establish the NN-RNALoc predictor through the following five steps:

1. Select or create a valid benchmark dataset for use in training and validating the predictor. This step is detailed in the "Data Sources" section.

2. Represent the mRNAs with an efficient formulation and extract k-mer information that reflects their intrinsic correlation with the to-be-predicted target. This process is outlined in "Feature Encoding" section.

3. Introduce and develop the potent NN-RNALoc algorithm for prediction purposes. In the "NN-RNALoc" section, the three primary steps of NN-RNALoc and its workflow are described.

4. Perform cross-validation tests to objectively evaluate the anticipated accuracy of the prediction. This section describes the third step of the NN-RNALoc workflow, which is also covered in the "NN-RNALoc" section.

5. In our future work, we will create a user-friendly, publicly accessible web server for the predictor. More information is provided in the "Conclusion" section.

The remaining sections are organized as follows: in the Materials and Methods section, we declare the datasets and introduce the features extracted from mRNA transcripts. Then, we discuss the specifics and steps required to create our model. In the Results section, we describe the performance of NN-RNALoc on the aforementioned two datasets and compare it to different methods: mRNALoc, RNATracker, DNN-5mer, DM3Loc, iLoc-mRNA, mLoc-mRNA, and EL-RMLocNet. In the Discussion section, we evaluate the performance of NN-RNALoc on human and non-human transcripts, utilizing novel distance-based subsequence profiles and canonical k-mer information.

## Materials and methods

This section describes the data sources utilized in our research. Then, the details of the features and the architecture of NN-RNALoc are explained in more depth.

### Data sources

**mRNA sequences and localization information.**   Two datasets are considered to benchmark the performance of NN-RNALoc against well-known algorithms. The initial dataset is CeFra-Seq, which is also utilized by the RNATracker technique. As stated previously, CeFra-Seq contains human transcripts, and localization information of mRNAs is presented as normalized gene expression values for each of four sub-cellular locations: Cytosol, Nucleus,

Membrane, and Insoluble. Therefore, rather than a single cellular location label for each mRNA, we have a four-element vector whose elements represent the probability of each mRNA's location. In this dataset, there are 11,373 mRNAs, and the sequences come from the Ensemble dataset [23]. For the second dataset (RNALocate), mRNA sequences and sub-cellular localization information are extracted from the RNALocate dataset. NN-RNALoc considers each human and non-human transcript separately, and for each gene, only one isoform is considered. This study examines the Cytoplasm, Endoplasmic Reticulum (ER), Extracellular Region (EX), Mitochondria, and Nucleus. In this dataset, only mRNA sequences belonging to a single location are taken into account. The RNALocate sub-cellular localization data were obtained from RNALocate at https://www.rna-society.org/rnalocate/. The sequences of mRNAs were downloaded from GenBank and the mRNA sequence data in the FASTA format were obtained from the NCBI on December 2022 [24]. In total, this dataset contains 11,180 mRNAs, of which 5,905 are human transcripts and 5,275 are non-human transcripts. Table 1 provides a summary of this dataset. Notably, because the data produced by APEX-RIP is fairly noisy [9, 12], we did not use it in this study.

**Protein-Protein Interaction (PPI) information.** The PPI information regarding human mRNA is extracted from the STRING database [25]. The longest protein-coding isoform among all isoforms of a gene is considered in this database, and one protein is then assigned to each mRNA. Thus, we obtain a weighted network for which vertices are the proteins assigned to mRNAs, the edges represent the interaction between the corresponding proteins, and the weights represent the STRING-assigned strength of the interaction between two proteins. So, the PPI information can be shown as a matrix for which entries show how strongly two proteins interact with each other.

## Feature encoding

With the explosive growth of biological sequences in the post-genomic era, one of the most important and challenging problems in computational biology is how to express a biological sequence using a discrete model or vector while retaining significant sequence-order information or essential pattern characteristics. Two types of characteristics are derived from mRNA sequences. The first is k-mer representation, one of the most commonly employed encodings for nucleotide sequences [26–28]. The second is a novel representation that we propose for mRNA sequences. These two characteristics are described in detail below.

**k-mer representation.** Counting k-mer frequencies is one way to extract a uniform-length feature vector from these sequences [26, 28]. A k-mer is a potential subsequence of length k within the mRNA sequence. As there are four neucleotides, there are a total of $4^k$

**Table 1. Total number of mRNAs in each five locations in the RNALocate dataset.**

| Location | Human Species | Non-human Species |
|---|---|---|
| Cytoplasm | 3,427 | 1,534 |
| Endoplasmic Reticulum | 1,173 | 8 |
| Extracellular Region | 26 | 509 |
| Mitochondria | 5 | 344 |
| Nuclear | 1,274 | 2,880 |
| Total | 5,905 | 5,275 |

The first column represents each cellular compartment. The second and third column reveal the number of human and non-human transcripts, respectively.

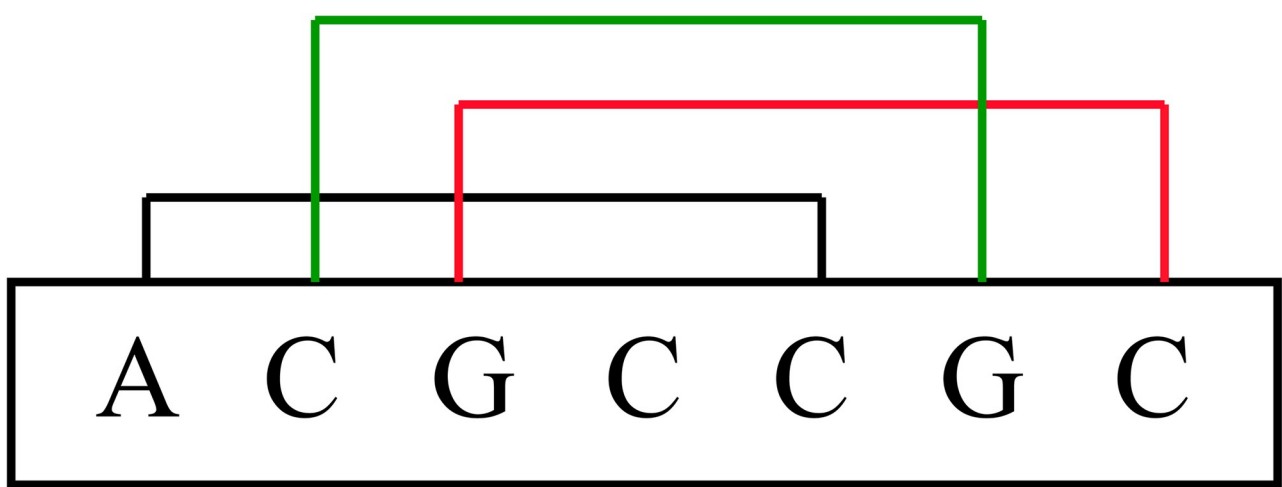

**Fig 1. All 5-mer structures contained in ACGCCGC sequence.** In this example, we have three 5-mers ACGCC, CGCCG and GCCGC that are shown in three different colors.

possible k-mers. Some k-mer profiles have been demonstrated to be more important for certain tasks. Specifically, Hart et al., discussed the significance of 5-mer sites in microRNA-gene targeting [29]. As shown in Fig 1, for the ACGCCGCCG sequence, all 5-mer structures are ACGCC, CGCCG, and GCCGC. The k-mer characteristics of this paper are covered by a highly effective web server called "Pse-in-One" [30]. For an mRNA sequence S, we sort all 5-mer structures lexicographically, then count the frequency of each 5-mer structure in the main mRNA sequence and divide it by its length. Consequently, we acquire the following attribute vector: $F_5(S) = [v_1, v_2, \ldots, v_n]$, where each $v_i$ is the frequency of the i-th 5-mer and n is equal to $4^5 = 1,024$ in this case.

**Distance-based sub-sequence profiles.** The main drawback of k-mer representation is that when k increases, the feature vector becomes extremely large and sparse, which can be memory-inefficient and can reduce the performance of the model. In order to mitigate the issue of small repeat regions, it may be advantageous to employ larger k-mer sizes. However, as the number of matching subsequences decreases, large k-mers become computationally infeasible and result in significant sparsity in the feature vector. In this study, we propose a novel distance-based representation to partially address this issue. In the novel distance-based profiles, the distance between the first and last nucleotide of the subsequence that we counted is k. The frequency of this subsequence is then determined for each pair of nucleotides separated by k. Consequently, for an mRNA sequence S and a distance k, the following 16-element feature vector is obtained: $D_k(S) = [w_1, w_2, \ldots, w_{16}]$, where $w_i$ is the frequency of each distance-based sub-sequence and X is a sub-sequence of size k. For any k, an illustration of all subsequences to count is provided in Fig 2.

It is obvious that for an mRNA sequence S with a length of m, X can be replaced with a sub-sequence of nucleotides (A, G, C, and T) ranging from size 0 to m-2. As an example, let's consider S to be the mRNA with the sequence ACGCCGC with a length of 7, so X can be a sub-sequence of maximum size 5. For example, in Fig 3, four distance-based substructures of ACGCCGC are shown in three different colors. The two sub-sequences CGCC and CCGC with distance 2 are shown in green, one sub-sequence GCCGC with distance 3 is drawn in red, and one sub-sequence ACGCCGC with distance 5 is illustrated in black. For instance, to calculate $w_3$ and $w_6$ in this sequence, for $w_3$: AXG, we have one sub-sequence ACG (k = 1) and one

$$w_1 : \text{AXA}, \quad w_2 : \text{AXC}, \quad w_3 : \text{AXG}, \quad w_4 : \text{AXT}$$
$$w_5. : \text{CXA}, \quad w_6 : \text{CXC}, \quad w_7 : \text{CXG}, \quad w_8 : \text{CXT}$$
$$w_9. : \text{GXA}, \quad w_{10}: \text{GXC}, \quad w_{11}: \text{GXG}, \quad w_{12}: \text{CXT}$$
$$w_{13}: \text{CXA}, \quad w_{14}: \text{CXC}, \quad w_{15}: \text{CXG}, \quad w_{16}: \text{CXT}$$

**Fig 2. For an mRNA sequence S and a distance k, we depict the 16-element feature vector, where $w_i$ is the frequency of each distance-based subsequence and X denotes a possible sub-sequences of size k.**

sub-sequence ACGCCG (k = 4), so the frequency of $w_3$ is 2. For $w_6$: CXC, the sequence contains one sub-sequence CC (k = 0), two sub-sequences CGC (k = 1), one sub-sequence CCGC (k = 2), one sub-sequence CGCC (k = 2), and one sub-sequence CGCCGC (k = 4). Therefore, the frequency of $w_6$ is 6. In this work, we tested a wide range of distances, and after many trials and errors, we found the best range for k to be between 0 and 8. As a result, the length of the created feature vector is $9 \times 16 = 144$.

**Principle Component Analysis on PPI network.** As previously stated, the PPI information is represented as an adjacency matrix with the dimension of number of mRNAs × number of mRNAs. As a result, the PPI matrix in the first dataset has a dimension of 11, 373 × 11, 373 whereas the PPI matrix in the RNALocate dataset has a dimension of 5880 × 5880. Because the performance of machine learning models can decrease when too many features are considered, we first employ the Principle Component Analysis (PCA) technique to reduce the dimension of this matrix [31]. PCA is one of the most widely used methods for reducing feature space and increasing storage space or the computational efficiency of a learning algorithm. It applies singular value decomposition to project data into a lower

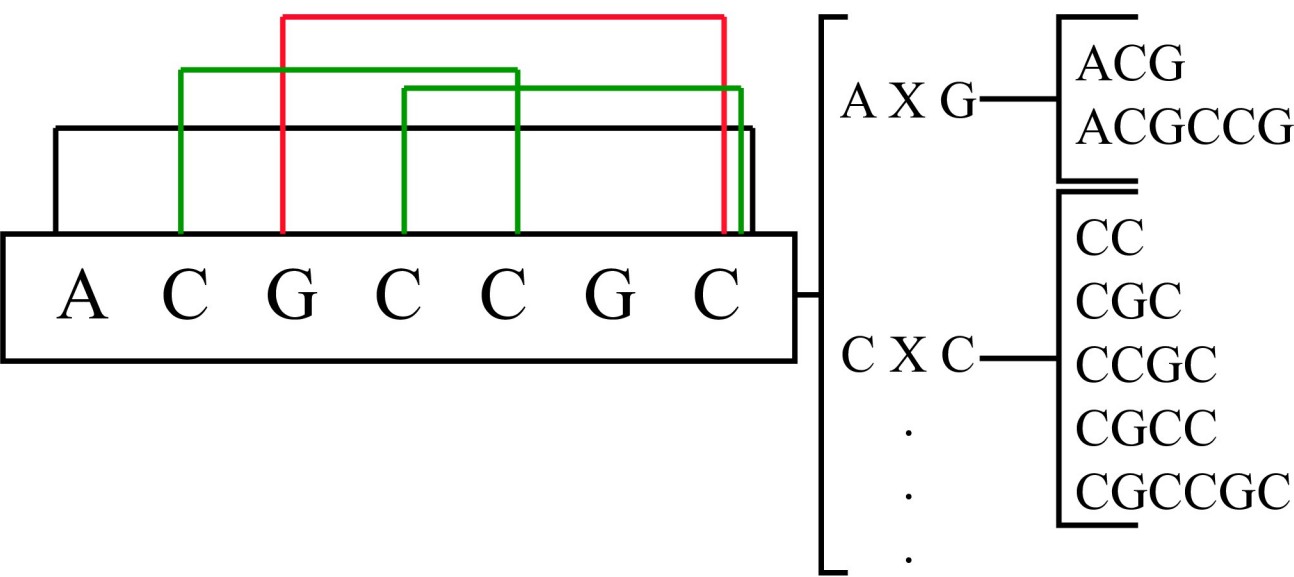

**Fig 3. Four distance-based substructures are shown in three different colors for the mRNA sequence S = ACGCCGC.** Two sub-sequences CGCC and CCGC with k = 2 are shown in green, one sub-sequence GCCGC with k = 3 is depicted in red, and one sub-sequence ACGCCGC with k = 5 is illustrated in black. In addition, the figure depicts the possible subsequences of S between A and G (AXG) and C and C (CXC).

dimensional space, emphasizing variation and highlighting strong patterns in a dataset. After many tests, the number of principal components in this study has been set to 500, and the total variance explained is more than 70% of the total data.

## NN-RNALoc

In this section, we express the main steps of NN-RNALoc.

**Step 1: Combine of the following three feature vectors:**

1. 5-mer frequencies (a vector of length 1,024)

2. Subsequence distance-based profiles (a vector of length 144)

3. Reduced PPI matrix using PCA method (a vector of length 500 for each mRNA)

We combine the collected data into a single 1668-dimensional feature vector (1,024 + 144 + 500). This vector serves as the final input for our neural network model for the prediction task.

**Step 2: Design of a neural network model**

We propose an artificial neural network (ANN) for assigning probabilities of a mRNA belonging to a specific location based on our developed features. A neural network can be represented as a sequence of matrix multiplications interleaved with nonlinear functions. An ANN is made up of a number of smaller units known as neurons, which can be repeated in multiple layers. To prevent the neural network from becoming complex and thus more difficult to train efficiently, we employ a model with a shallow architecture consisting of one hidden layer and 200 neurons. Dropout is also used in the hidden layer to mask randomly 50% of the connections during model training to prevent overfitting. We use the Rectified Linear Unit (Relu) activation function in the hidden layer, which is described as follows [32]:

$$\text{Relu}(x) = max(0, x). \tag{1}$$

The Softmax function as the non-linear function is applied in the last layer of the model to assign a probability to each location ($x_i$) and is formulated as bellow [33]:

$$\text{Softmax}(x_i) = \frac{\exp(x_i)}{\sum_j \exp(x_j)} \tag{2}$$

Finaly, we use Kullback-Leiber-Divergence as the loss function. For probability distribution $\mathfrak{P}$ and $\mathfrak{Q}$ defined on the same probability space X, Kullback-Leiber-Divergence is defined as [34]:

$$KL(\mathfrak{Q} \parallel \mathfrak{P}) = \Sigma_{x \in X} \mathfrak{Q}(x) \left( log \frac{\mathfrak{Q}(x)}{\mathfrak{P}(x)} \right). \tag{3}$$

**Step 3: Training of the prediction model**

The selection of hyper-parameters of the model is based on the training dataset. All parameters were chosen with the intent of minimizing the loss function. For training the model, we employ the 10-fold cross-validation method [35]. The outcomes are then evaluated using a range of values for hidden layers (no hidden layer, 1, 2, and 3), neurons in each fully connected layer (1,000, 700, 500, 200, and 100), and dropout rates (0.1, 0.2, 0.3 and 0.5). Table 2 displays the most optimal parameters utilized by this model. A validation set consisting of 10% of the training data is also applied to monitor the loss function during the training process and detect overfitting. The Keras Library [36] is used to implement NN-RNALoc. In addition, the Adam

**Table 2. Selected hyper-parameters for NN-RNALoc.**

| Parameter | Value |
| --- | --- |
| Epochs | 300 |
| Batch size | 512 |
| Hidden layer | 1 |
| Fully connected neurons | 200 |
| Dropout rate | 0.2 |

optimizer with Nesterov momentum is used to train the model [37]. Fig 4 depicts the comprehensive workflow of NN-RNALoc.

## Evaluation criteria

As stated previously, we work with two datasets, and due to the differences in their structures, we compare different metrics to evaluate the performance of the model on each dataset. As described previously, the CeFra-Seq localization values are continuous. We therefore consider correlation measurements when evaluating model performance similar to [9] study. The initial measure is Pearson Correlation. Pearson Correlation is a method for measuring the linear

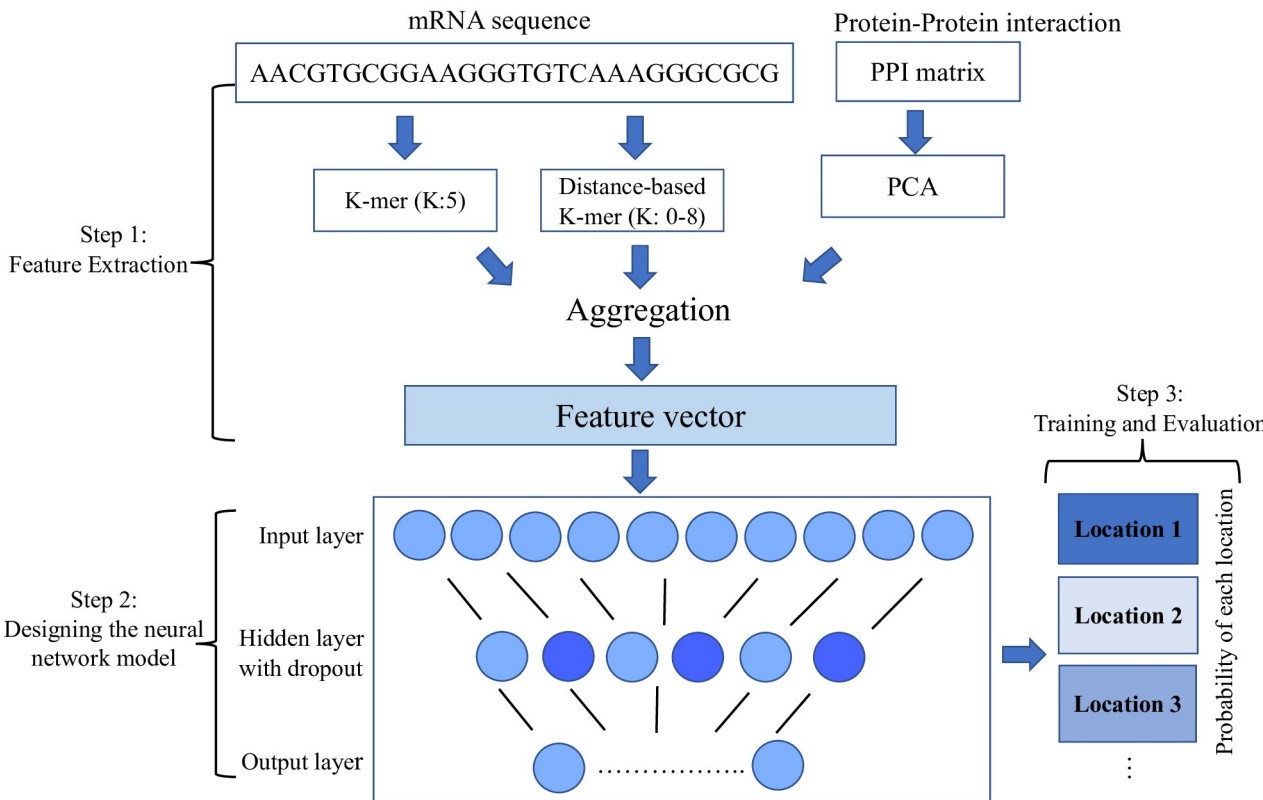

**Fig 4. The overview of NN-RNALoc method.** In Step 1, we first aggregate all information gathered from both sequence-based features as well as protein-protein interaction (PPI) matrix. In Step 2, we design a neural network model with the mentioned architecture. In Step 3, the model is trained and evaluated using 10-fold cross-validation. We report a probability vector of length 4 for each mRNA in the CeFra-Seq dataset and the location with the highest probability as the mRNA's predicted location in the RNALocate dataset.

correlation between predicted and observed values. It has a value between 1 and -1, with +1 representing a total positive linear correlation, 0 representing no linear correlation, and -1 representing a total negative linear correlation. In order to better evaluate the performance of the model, we also consider the Spearman correlation between predicted and experimental values to capture the order of locations to which an mRNA belongs. In addition, we employ classification metrics in the RNALocate dataset because localization information is discrete values similar to [12] study. True Positive (TP), True Negative (TN), False Positive (FP), False Negative (FN), Precision, Recall, F-score, Accuracy (ACC), and Matthews Correlation Coefficient (MCC) are computed for this dataset in order to compare the performance of the NN-RNALoc method to that of other methods. These criteria are defined below:

$$
\begin{aligned}
Precision &= \frac{TP}{TP+FP}, \ Recall = \frac{TP}{TP+FN}, \\
F-score &= 2\frac{Recall \times Precision}{Recall + Precision}, \\
ACC &= \frac{TP+TN}{TP+FP+FN+TN}, \\
MCC &= \frac{TP \times TN - FP \times FN}{\sqrt{(TP+FP)(TP+FN)(TN+FP)(TN+FN)}}
\end{aligned}
$$

The criteria listed above are some of the most prevalent metrics used in classification problems to evaluate the performance of a model. Clearly, relying solely on a single statistical measure (such as ACC) can lead to overoptimistic results, particularly when analyzing datasets with imbalances. As a result, we also evaluate enhancements to the MCC measure, which is a more reliable statistical rate that yields a high score only if the predictive model achieves success in all categories. In Table 3, we have summarized all the metrics used to evaluate the performance of the models on the two benchmarks.

## Results

### Time complexity

Many of the methodologies, including mRNALoc, are provided as server-based tools, making it impossible to compare their time complexity. But the source code and implementation of RNATracker are available, so we can compare how long it takes to run RNATracker, DNN-5mer, and NN-RNALoc. On a Linux Ubuntu machine with 15 CPUs (Intel Xeon(R) 2.00 GHz) and the CeFra-Seq dataset, NN-RNALoc takes approximately 3 hours, which is significantly faster than RNATracker (RNATracker requires 7 days for training in full length mode and 8 hours in fixed length mode on GTX1080Ti graphic card). On the RNALocate dataset,

**Table 3. A summary of the localization information for two datasets and metrics used to assess models performance.**

| Dataset | Localization information | Metrics |
|---|---|---|
| CefraSeq | Normalized gene expression valuess | Regression metrics: Pearson Cor. and Spearman Cor |
| RNALocate | Single location | Classification metrics: Precision, ACC, F-score, MCC |

We use correlation measurements for CefraSeq, and classification metrics for RNALocate dataset. Pearson Cor: Pearson Correlation; Spearman Cor: Spearman Correlation; ACC: Accuracy; MCC: Matthews Correlation Coefficient.

NN-RNALoc's time is 2 hours, which is significantly less than RNATracker's computation time requirement of 6 hours. On the other hand, the computational time of DNN-5mer is comparable to that of NN-RNALoc. Consequently, in terms of training time, we can conclude that NN-RNALoc significantly outperforms RNATracker, while having the same time complexity as DNN-5mer.

## Assessment and comparison

The results of 30 times of 10-fold cross-validation on the CeFra-Seq dataset are displayed in Table 4. In the Cytosol, Insoluble, Membrane, and Nuclear regions of the CeFra-Seq dataset, NN-RNALoc finds Pearson correlations of 0.69, 0.65, 0.54, and 0.55, respectively. In every location, these correlations are stronger when compared to the RNATracker fixed length mode method. Using NN-RNALoc, the number of mRNAs with a Spearman correlation of 1, indicating a perfect association of ranks, is 2893, which is slightly better than RNATracker (2849). As demonstrated in Table 4, NN-RNALoc achieves approximately 17% greater overall Pearson correlation than RNATracker's fixed length mode. Due to the fact that mRNALoc is a standalone tool trained on five different locations than CeFra-Seq, it was not possible to compare it to NN-RNALoc for this dataset. DNN-kMer is a multilayer perceptron-based predictor that extracts k-mer features from sequences (1-mers to k-mers). In both data sets, the DNN-kMer model was trained on 1-mers to 8-mers, and the best results were obtained when all 1-mer to 5-mer information was taken into account. Therefore, DNN-5mer's inputs are a 1364-dimensional ($4^1 + 4^2 + 4^3 + 4^4 + 4^5$) vector. As a result, using 1-mers to 5-mers as features, we evaluate the performance of NN-RNALoc and DNN-5mer. DNN-5mer has only two hidden layers with the same number of neurons as the input vector. In the hidden layer, the Relu activation function is utilized. Despite the fact that both NN-RNALoc and DNN-5mer have a simple architecture, DNN-5mer performs significantly worse, with Pearson correlations of 0.63 in the Membrane, 0.55 in Insoluble, 0.42 in the Membrane, and 0.48 in the nucleus. Overall, NN-RNALoc achieves a Pearson correlation approximately 35% higher than DNN-5mer. In addition, we ran NN-RNALoc with only k-mer frequencies (for k from 1 to 5) to evaluate the effect of incorporating the distance-based profile into the model. As Table 4 represents, in this context (comparing NN-RNALoc(no PPI) and NN-RNALoc(k-mer profile)) the Pearson correlations were 9% lower in total, demonstrating the advantages of using distance-based profiles.

RNALocate is the most well-known dataset in this field and was used for validation for all the algorithms mentioned in the previous studies. The performance of NN-RNALoc on RNALocate is benchmarked against RNATracker, DM3Loc, mRNALoc, iLoc-mRNA, EL-RMLocNet, and mLoc-mRNA methods. We report the area under the Receiver Operator Characteristic (ROC) curve (AUC-ROC) and the area under the Precision-Recall (PR)

**Table 4. Average Pearson correlations of 30 times 10-fold cross-validation in each location of Cefra-Seq dataset obtained by different methods.**

| Location | NN-RNALoc | NN-RNALoc(no PPI) | NN-RNALoc(k-mer profile) | RNATracker fixed | RNATracker full | DNN-5mer |
|---|---|---|---|---|---|---|
| Cytosol | 0.69 | 0.67 | 0.66 | 0.68 | **0.70** | 0.63 |
| Insoluble | **0.65** | 0.61 | 0.60 | 0.62 | 0.64 | 0.55 |
| Membrane | **0.54** | 0.52 | 0.47 | 0.47 | **0.54** | 0.42 |
| Nuclear | **0.55** | 0.52 | 0.50 | 0.49 | 0.54 | 0.48 |

NN-RNALoc (with employing PPI, k-mer and distance-based profiles); NN-RNALoc(no PPI) (k-mer and distance-based profiles); NN-RNALoc(only k-mer); RNATracker (fixed length mode); RNATracker full (full length mode);DNN-5mer (1-mers to 5-mers)

curve (AUC-PR) for a fair comparison of the tested methods similar to RNATracker, DM3Loc, mRNALoc, iLoc-mRNA, EL-RMLocNet, and mLoc-mRNA studies. Table 5 summarizes the AUC-ROC, AUC-PR, and Average MCC for different methods for the human part of the RNALocate dataset. For Cyt location, NN-RNALoc and mRNALoc outperformed others based on AUC-ROC and AUC-PR, respectively. For ER, iLoc-mRNA and NN-RNALoc outperformed others based on AUC-ROC and AUC-PR, respectively. For EX, mLoc-mRNA and RNATracker outperformed others based on AUC-ROC and AUC-PR, respectively. For the Nuc location, mLoc-mRNA and RNATracker outperformed others based on AUC-ROC and AUC-PR, respectively. As seen in Table 5, none of the methods outperform the other methods in all locations and for Cyt and ER locations, NN-RNALoc outperformed well-known methods. Similar to some previous methods, we only considered single-location mRNA sequences in the RNALocate dataset. Except for DM3Lo and mLocmRNA methods, which predict multiple locations for each mRNA sequence, all other methods only predict a single location. If the actual location of an mRNA sequence was presented in the prediction results of the mLocmRNA and DM3Lo methods, and it was reported as a true prediction. It is obvious that by predicting multiple locations, these methods improve the performance of their algorithm in some locations compared to other methods, as shown in Table 5. Similarly, Table 6 represents the result of different methods on the non-human part of the RNALocate dataset. In this case, NN-RNALoc outperformed existing methods for the Nuc location and obtained nearly similar results to other methods. In terms of average MCC, NN-RNALoc performs better than other methods, which shows that our method works well overall.

The performance of NN-RNALoc's on the RNALocate dataset for the best threshold has been reported as follows: The Precision, Recall, and F-score values for Cytosol using NN-RNALoc are 74%, 72%, and 74%, respectively. Endoplasmic Reticulum (ER) has a precision of 56%, a recall of 48%, and an F-score of 52%. In the Extracellular Region (EX) and Mitochondria, due to a lack of training samples (only 26 and 2, respectively), the Recall and F-score are close to zero. The precision of prediction in nucleus is 52%, whereas recall and F-score are 70% and 60%, respectively. In fact, NN-RNALoc increased the total F-score in all locations by about 17% compared to mRNALoc and by 56% compared to RNATracker. However, in the nucleus, the average F-score obtained for NN-RNALoc and mRNALoc is nearly identical. The overall accuracy of prediction using NN-RNALoc is higher than both RNATracker and mRNALoc. NN-RNALoc additionally achieves an MCC of 0.40, which is greater than RNATracker and mRNALoc (they both achieve an MCC of 0.34 and 0.37, respectively). Results have been shown in S1 Table.

In addition, we used other shallow learning algorithms e.g. SVM, RF, Extreme Gradient Boosting (XGBoost), and light gradient-boosting machine (LGBM) [38] for our learning process methods instead of using NN. SVM-RNALoc used SVM on k-mer and distance-based profile features, XGBoost-RNALoc employed XGBoost on k-mer and distance-based profile features, and LightGBM-RNALoc applied LightGBM on k-mer and distance-based profile features. Table 7 and S2 Table indicate the results of these algorithms for the Cefra-Seq and RNALocate datasets, respectively. The results show that NN-RNALoc for most locations outperforms other methods. Hence, we used the NN method to predict locations based on k-mer and distance-based profile features. Moreover, we applied the DNN-kMer method which is a multilayer perceptron-based predictor that extracts k-mer features from sequences (1-mers to k-mers) and compared them with NN-RNALoc (please see Table 4). The results show that NN-RNALoc outperforms the other shallow learning approaches.

**Table 5. Results of AUC-ROC and AUC-PR for different methods on the human part of the RNALocate dataset.**

| Method | Compartment | AUC-ROC | AUC-PR | Average MCC |
|---|---|---|---|---|
| NN-RNALoc | Cyt | **0.76** | 0.71 | **0.40** |
| | ER | 0.71 | **0.79** | |
| | EX | 0.65 | 0.63 | |
| | Mit | 0 | 0 | |
| | Nuc | 0.79 | 0.77 | |
| NN-RNALoc (noPPI) | Cyt | 0.73 | 0.67 | 0.30 |
| | ER | 0.66 | 0.55 | |
| | EX | 0 | 0 | |
| | Mit | 0 | 0 | |
| | Nuc | 0.70 | 0.74 | |
| RNATracker | Cyt | 0.73 | 0.31 | 0.34 |
| | ER | 0.62 | 0.18 | |
| | EX | 0.75 | **0.99** | |
| | Mit | 0 | 0 | |
| | Nuc | 0.75 | **0.86** | |
| DM3Loc | Cyt | 0.74 | 0.31 | 0.24 |
| | ER | 0.69 | 0.25 | |
| | EX | 0 | 0 | |
| | Mit | 0 | 0 | |
| | Nuc | 0.77 | 0.87 | |
| mRNALoc | Cyt | 0.60 | **0.76** | 0.37 |
| | ER | 0.37 | 0.14 | |
| | EX | 0.40 | 0.98 | |
| | Mit | 0 | 0 | |
| | Nuc | 0.60 | 0.76 | |
| iLoc-mRNA | Cyt | 0.51 | 0.72 | 0.20 |
| | ER | **0.81** | 0.57 | |
| | EX | 0 | 0 | |
| | Mit | 0 | 0 | |
| | Nuc | 0.51 | 0.72 | |
| EL-RMLocNet | Cyt | 0.74 | 0.45 | 0.38 |
| | ER | 0 | 0 | |
| | EX | 0.75 | 0.67 | |
| | Mit | 0 | 0 | |
| | Nuc | 0.68 | 0.56 | |
| mLoc-mRNA | Cyt | 0.75 | 0.71 | 0.38 |
| | ER | 0.75 | 0.72 | |
| | EX | **0.76** | 0.77 | |
| | Mit | 0.98 | 0.99 | |
| | Nuc | **0.80** | 0.79 | |

## Discussion

To evaluate the effect of incorporating PPI information into our model, the following analysis was performed on both the CeFra-Seq and RNALocate datasets. We only utilize 5-mer and also distance-based sub-sequence information derived from mRNA sequences and compare the results to the scenario in which PPI information is also incorporated into the model. When

**Table 6. Results of AUC-ROC and AUC-PR for different methods on the non-human part of the RNALocate dataset.**

| Method | NN-RNALoc | | | | | RNATracker | | | | | mRNALoc | | | | | iLoc-mRNA | | | | | EL-RMLocNet | | | | |
|---|---|---|---|---|---|---|---|---|---|---|---|---|---|---|---|---|---|---|---|---|---|---|---|---|---|
| Compartment | Cyt | ER | EX | Mit | Nuc | Cyt | ER | EX | Mit | Nuc | Cyt | ER | EX | Mit | Nuc | Cyt | ER | EX | Mit | Nuc | Cyt | ER | EX | Mit | Nuc |
| AUC-ROC | 0.71 | 0 | 0.4 | 0.71 | 0.54 | **0.77** | 0 | 0.45 | **0.9** | 0.68 | 0.71 | 0.63 | **0.48** | 0.76 | 0.44 | 0.23 | **0.65** | 0 | 0 | **0.69** | 0.73 | 0 | 0 | 0.7 | 0.78 |
| AUC-PR | 0.77 | 0 | 0.38 | 0.93 | **0.72** | 0.69 | 0 | **0.5** | 0.85 | 0.7 | 0.57 | 0.1 | 0.23 | **0.99** | 0.71 | 0.16 | 0.48 | 0 | 0 | 0.56 | **0.8** | 0 | 0 | 0.59 | 0.68 |
| MCC | **0.55** | | | | | 0.43 | | | | | 0.47 | | | | | 0.38 | | | | | 0.5 | | | | |

The names of compartments are abbreviated as Cyt: Cytosol, ER: Endoplasmic Reticulum, EX: Extracellular Region, Mit:Mitochondria, Nuc: Nucleus.

**Table 7. Average Pearson correlations of 30 times 10-fold cross-validation in each location of Cefra-Seq dataset obtained by NN-RNALoc, SVM-RNALoc, RF-RNA-Loc, XGBoost-RNALoc, DNN-RNALoc, LGBM-RNALoc.**

| Location | NN-RNALoc | SVM-RNALoc | RF-RNALoc | XGBoost-RNALoc | LGBM-RNALoc |
|---|---|---|---|---|---|
| Cytosol | 0.69 | 0.65 | **0.77** | 0.45 | 0.65 |
| Insoluble | **0.65** | 0.43 | 0.37 | 0.56 | 0.33 |
| Membrane | **0.54** | 0.33 | 0.45 | 0.36 | 0.45 |
| Nuclear | **0.52** | 0.35 | 0.43 | 0.47 | 0.42 |

NN-RNALoc (with employing NN on k-mer and distance-based profiles features); SVM-RNALoc (with employing support vector machine on k-mer and distance-based profiles features); XGBoost-RNALoc(with employing extreme gradient boosting on k-mer and distance-based profiles features); LGBM-RNALoc (with employing light gradient-boosting machine on k-mer and distance-based profiles features).

the reduced PPI matrix is used in the model for the Cefra-se dataset, NN-RNALoc achieves almost 11% higher Pearson correlation in total for all locations, as shown in Table 4. We conduct the same analysis on the RNALocate dataset and human-related transcripts too, utilizing only sequence-based information in the model. These results, which are the same as those found in the first dataset, also show that when NN-RNALoc uses PPI information in the second dataset, its performance totally improves with 10% increase in MCC and 2% in accuracy. Fig 5 compiles the results for a more precise comparison of the performance of the NN-RNA-Loc algorithm with PPI information (NN-RNALoc) and without PPI information (NN-RNA-Loc(no PPI)) besides other methods. Fig 5(a) displays the resulted average of Pearson correlation for the CeFra-Seq dataset for four locations, and Fig 5(b) shows the average of F-score values for the five locations in the RNALocate dataset. According to Fig 5, considering PPI information improves the results for all locations in both datasets and has the greatest influence on predicting the insoluble location in CeFra-Seq dataset and Endoplasmic Reticulum location in the RNALocate dataset. To evaluate the impact of including distance-based profiles in the model, we omit this information from the feature vector. As previously discussed in the results and as shown in Table 4 and Fig 5, the poorer performance of NN-RNA-Loc on both datasets when only k-mer frequencies (for k from 1 to 5) are used can potentially demonstrate the impact of distance-based profiles. We then examine the non-human transcripts within the RNALocate dataset. Due to the large number of species whose transcripts are included in the dataset, the PPI information cannot be used in this instance. Consequently, only 5-mer and distance-based subsequence profiles of mRNA sequences are utilized in the model. Table 5 compares the performance of NN-RNALoc, mRNALoc, and RNATracker on non-human species. In this instance, the total accuracy obtained by NN-RNALoc is 74% which is 4% higher than RNATracker and 9% higher than mRNALoc. Moroevr, in this dataset, NN-RNALoc achieves MCC of 55% which is 8% higher than RNATracker and 12% higher

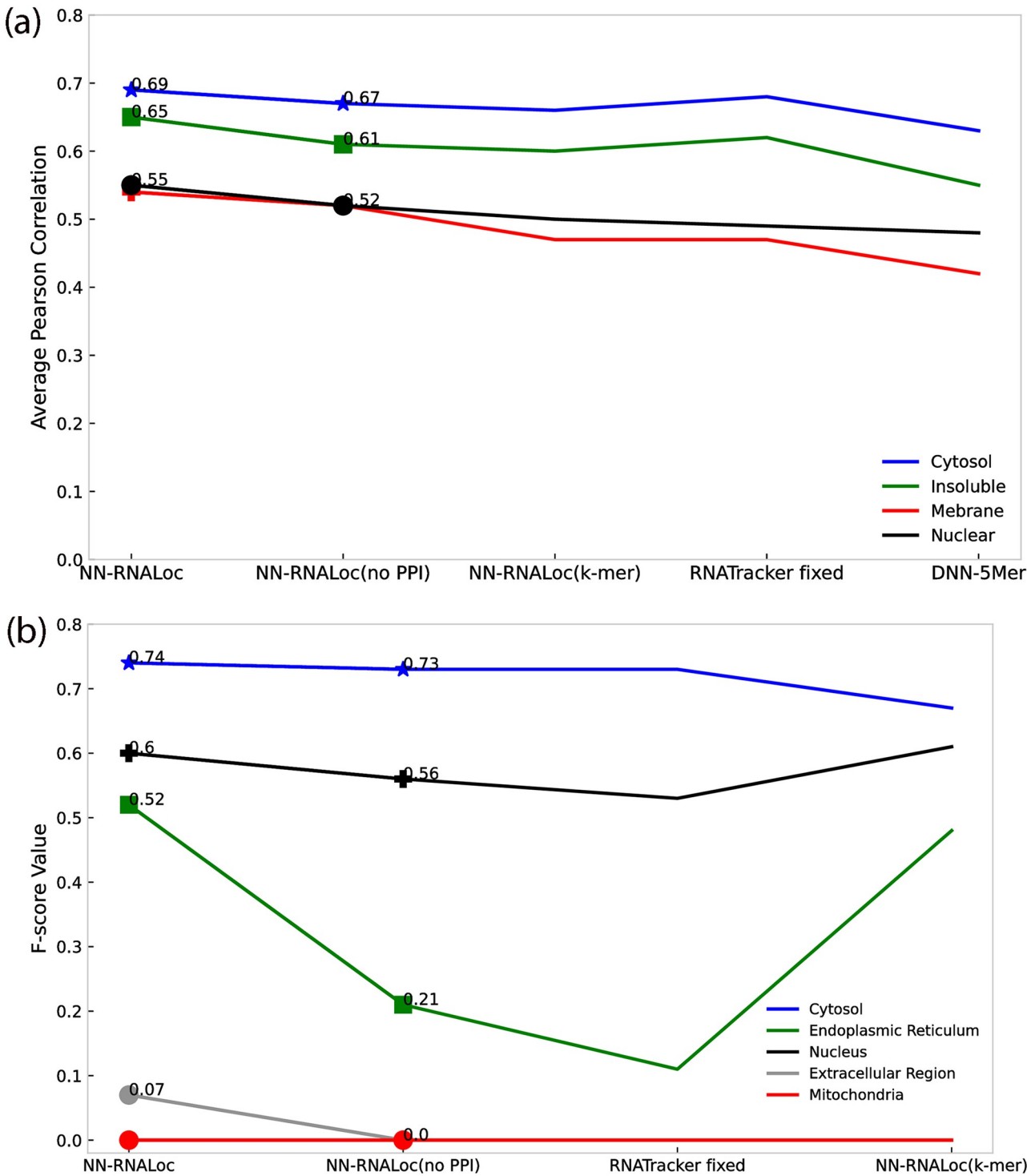

**Fig 5. Comparison of Pearson correlations and F-measure values of NN-RNALoc algorithm with other methods for two datasets.** (a) The average of Pearson correlation for the CeFra-Seq dataset for four locations. (b) The average of F-score values for the five locations in the RNALocate dataset.

than mRNALoc. Finally, for a more detailed evaluation and to determine the impact of each distance-based k-mer on the prediction of mRNA location, the following experiment was conducted on the CeFra-Seq dataset. We independently considered each distance-based profile for k ranging from 0 to 8. Fig 6 depicts the average Pearson correlation in each of four locations when a single distance-based k-mer profile was used. Using 8-mer distance-based profiles yields the highest correlation in Cytosol, Insoluble, and Nuclear, which are represented by blue, orange, and green curves, respectively, as shown in Fig 6. However, for Membrane, which is depicted by a red curve, the highest correlation is obtained using a 4-mer distance-based profile, despite the fact that the differences in Pearson correlations are negligible. Therefore, in order to find all possible patterns in mRNA sequences, we decided to look at the combination of distance-based profiles for all k-mers in the range of 0 to 8.

Our method has been evaluated using two different datasets. The first dataset, CeFra-Seq, uses a continuous set of values to represent the localization probability of each of the four compartments. Hence, we predict a probability value for each compartment of this dataset. Then, we use Pearson and Spearman correlations to assess the performance of the models in the CeFra-Seq dataset. Using our method, we can either select one location using the maximum probability value or select multiple locations by setting a probability threshold. The second dataset, compiled from the RNALocate dataset, is among the most commonly used datasets for RNA localization and all methods applied for the comparison report their results on this dataset. The element information of this dataset is a binary vector indicating whether a specific

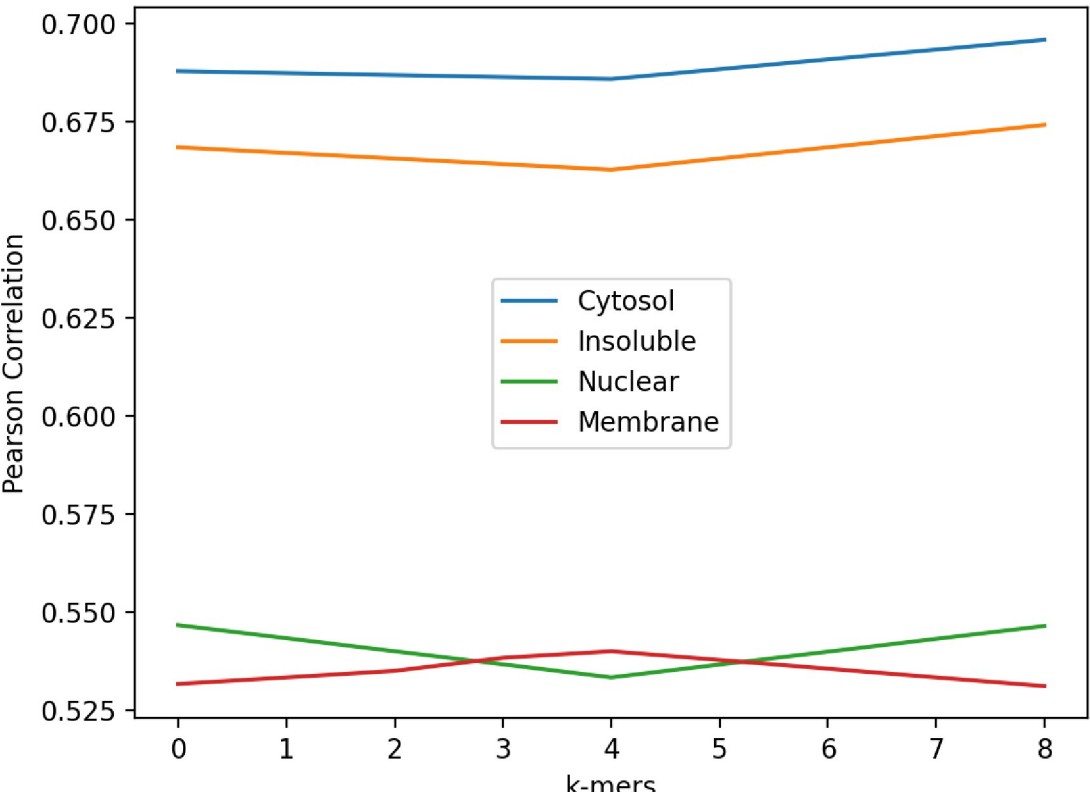

**Fig 6. Pearson correlation obtained by NN-RNALoc on CeFra-Seq dataset when employing each distance-based profile for k in range 0 and 8, individually.** Four locations are represented in four different colors; blue: Cytosol, orange: Insoluble, green: Nuclear, red: Membrane.

**Table 8. Performance of NN-RNALoc, RNATracker(fixed length mode) and mRNALoc on non-human mRNAs of RNALocate dataset.**

| Methods | NN-RNALoc | | | | | mRNALoc | | | | | RNATracker | | | | |
|---|---|---|---|---|---|---|---|---|---|---|---|---|---|---|---|
| Criteria | Cyt | ER | EX | Mit | Nuc | Cyt | ER | EX | Mit | Nuc | Cyt | ER | EX | Mit | Nuc |
| Precision | **0.77** | 0.00 | **0.38** | 0.98 | 0.72 | 0.57 | **0.01** | 0.23 | **0.99** | **0.77** | 0.69 | 0.00 | 0.50 | 0.85 | 0.70 |
| Recall | 0.54 | 0.00 | 0.03 | **0.98** | **0.96** | **0.57** | **0.33** | **0.18** | 0.85 | 0.76 | 0.54 | 0.00 | 0.001 | 0.75 | 0.92 |
| F-score | **0.63** | 0.00 | 0.05 | **0.98** | **0.82** | 0.57 | **0.02** | **0.20** | 0.92 | 0.77 | 0.60 | 0.00 | 0.003 | 0.79 | 0.79 |
| Average Accuracy | **0.74** | | | | | 0.65 | | | | | 0.70 | | | | |
| Average MCC | **0.55** | | | | | 0.43 | | | | | 0.47 | | | | |

The names of compartments are abbreviated as Cyt: Cytosol, ER: Endoplasmic Reticulum, EX: Extracellular Region, Mit:Mitochondria, Nuc: Nucleus.

RNA is present at a given location or not. Given that this dataset contains five locations, the length of this binary vector is also five. We use a classification method on this dataset to predict the localization of a given mRNA. For evaluating the performance of the classification algorithms, precision, recall, f-score, MCC, and ACC were used. We also reported AUC-ROC and AUC-PR for classification performance comparisons. It is crucial to note that for NN-RNALoc, the probability of each location for each mRNA is computed, then sorted, and the location with the highest probability is reported as the specific mRNA location. To assign more than one location to an mRNA, a threshold can be considered, and all locations with probabilities greater than the chosen threshold can be assigned to the mRNA sequences. However, in order to compare the results of this method with those of other methods, we assign the most probable location. It is worth mentioning that while there is no approach that outperforms the others for predicting all locations, we intend to integrate several methods to predict locations based on a voting measure in our future study.

## Conclusion

NN-RNALoc is one of the few methods proposed that uses neural network-based approaches to examine the cellular localization of mRNAs. As a result of the explosive growth of biological sequences discovered in the post-genomic era, and in order to use them in a timely manner for a variety of bioinformatics problems such as RNA and protein localization or drug development, a significant amount of sequence-based information, such as PTM (posttranslational modification) sites in proteins, has been successfully predicted [39]. The rapid development of sequential bioinformatics and structural bioinformatics, as well as the introduction of computational methodologies for this purpose, have led to an unprecedented revolution in this field of study. Consequently, computational (or in silico) methods were also utilized in this study. Localization of messenger RNA (mRNA) molecules within the Cytoplasm provides a foundation for cell polarization, thereby underpinning developmental processes such as asymmetric cell division, cell migration, neuronal maturation, and embryonic patterning [40]. The enormous benefit of mRNA targeting is that it allows for the regulation of gene expression in both space and time; thus, RNA localization would be beneficial for understanding cellular functions [40]. NN-RNALoc is a neural network-based tool that aims to predict the subcellular localization of mRNA based on the interaction information of the proteins encoded by the mRNA transcripts. In this way, we have come up with a different distance-based subsequence profile for representing mRNA sequences. This novel encoding, which is more compact and less likely to add redundant data, was created to address the memory and time issues that arise as k in k-mer representation increases. Using distance-based sub-sequence profiles, k-mer frequencies, and reduced PPI matrix data, the results demonstrate that NN-RNALoc, a neural

network with a simple and transparent architecture, outperforms three previously introduced and powerful methods. This simplicity also drastically reduces the computation time required for model training. The application of a dimensional reduction technique, such as PCA, to the PPI data, which is a high-dimensional matrix, is significantly more advantageous than the use of raw interaction patterns. In the future, additional dimension reduction techniques, such as auto-encoders and PPI-specific compression techniques, can be investigated. In addition, it is important to note that future research can utilize the incorporation of other important but difficult-to-implement features, such as the knowledge of protein 3D structures or their complexes with ligands, which is crucial in numerous studies such as drug design [41]. Therefore, in future versions of NN-RNALoc, the incorporation of protein structural information could also be investigated. Moreover, as demonstrated by a number of recent publications [42, 43] demonstrating new findings or approaches, user-friendly and publicly accessible web-servers will significantly increase their impacts [20, 21]. So, in our future work, we will try to make a web server that can be changed by the user and show the results.

## Supporting information

**S1 Table. Results of 30 times 10-fold cross-validation of NN-RNALoc (with and without employing PPI information) compared with RNATracker(fixed length mode) and mRNA-Loc on the human part of RNALocate dataset.**
(PDF)

**S2 Table. Results of AUC-ROC (ROC) and AUC-PR (PR) for each location of human part of RNALocate database obtained by NN-RNALoc, SVM-RNALoc, RF-RNALoc, XGBoost-RNALoc, DNN-RNALoc, LightGBM-RNALoc.**
(PDF)

## Acknowledgments

Changiz Eslahchi and others would like to thank the School of Biological Sciences, Institute for Research in Fundamental Sciences (IPM) and Computing Center of IPM in performing a parallel computing is gratefully acknowledged.

## Author Contributions

**Conceptualization:** Negin Sadat Babaiha.

**Data curation:** Negin Sadat Babaiha.

**Formal analysis:** Negin Sadat Babaiha, Shokoofeh Ghiam, Changiz Eslahchi.

**Investigation:** Negin Sadat Babaiha, Changiz Eslahchi.

**Methodology:** Negin Sadat Babaiha, Changiz Eslahchi.

**Project administration:** Negin Sadat Babaiha.

**Resources:** Negin Sadat Babaiha.

**Software:** Negin Sadat Babaiha.

**Supervision:** Rosa Aghdam, Changiz Eslahchi.

**Validation:** Negin Sadat Babaiha, Rosa Aghdam.

**Writing – original draft:** Negin Sadat Babaiha.

**Writing – review & editing:** Negin Sadat Babaiha, Rosa Aghdam, Shokoofeh Ghiam, Changiz Eslahchi.

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
