## [Decision Letter · Decision Letter 0]

20 Apr 2022

PONE-D-21-31782NN-RNALoc: neural network-based model for prediction of mRNA sub-cellular localization using distance-based sub-sequence profilesPLOS ONE

Dear Dr. Eslahchi,

Thank you for submitting your manuscript to PLOS ONE. After careful consideration, we feel that it has merit but does not fully meet PLOS ONE’s publication criteria as it currently stands. Therefore, we invite you to submit a revised version of the manuscript that addresses the points raised during the review process.

We look forward to receiving your revised manuscript.

Kind regards,

Jianhong Zhou

Staff Editor

PLOS ONE

2. Please amend either the abstract on the online submission form (via Edit Submission) or the abstract in the manuscript so that they are identical.

Reviewers' comments:

Reviewer's Responses to Questions

**Comments to the Author**

1. Is the manuscript technically sound, and do the data support the conclusions?

Reviewer #1: Yes

Reviewer #2: Yes

2. Has the statistical analysis been performed appropriately and rigorously? 

Reviewer #1: Yes

Reviewer #2: Yes

3. Have the authors made all data underlying the findings in their manuscript fully available?

Reviewer #1: Yes

Reviewer #2: Yes

4. Is the manuscript presented in an intelligible fashion and written in standard English?

Reviewer #1: No

Reviewer #2: Yes

5. Review Comments to the Author

Reviewer #1: NN-RNALoc: neural network-based model for prediction of mRNA sub-cellular localization using distance-based sub-sequence profiles

Summary:

The authors present NN-RNALoc a new machine learning classifier for the prediction of mRNA subcellular localization. While there have been multiple previously published classification models for this task, the authors incorporate two novel sets of features, protein-protein interaction networks and distance-based subsequence profiles. The incorporation of these new features results in small improvements over previous methods. The improvements in classifier accuracy, while small, are noteworthy accomplishments. However, one key area of improvement for this manuscript would be to show some biological insights their new model can identify that previous classifier cannot. As it stands now, all of the figures in the results section are benchmarking tables. Benchmarks are crucial when developing a new model but so are biological insights, which this paper is currently lacking.

Comments:

1. The text has many grammatical and spelling errors, which need to be addressed to be able to better appreciate the work presented. For instance, “dimension” and “distance” are misspelled on line 57 and the Figure 2 legend, respectively.

2. The authors spend a considerably large amount of text on the introduction which spans, 144 lines of text, over 2 pages even enumerating machine learning guidelines from a previous study. I would recommend the introduction be condensed into more concise text which would make the transition to methods and results smoother.

3. Why is a distance-based sub-sequence of k=2 optimal, why not larger values? It seems like k=2 is capturing information already present in the k-mers counts and would be interesting to hear the authors discuss their methodology for selecting k=2.

4. In table 3 and 4, two benchmarks are performed, however, the authors utilize two different metrics for evaluation. Table 3 is correlation based while table 4 uses the standard multi-class accuracy metrics. This is slightly confusing because they are all performing the same classification tasks, the metrics used should be the same between benchmarks to enable better comparisons.

5. The authors state the advantages of their distance-based sub-sequence profiles many times but do not directly quantify their benefits. It would be informative for the authors to create a new model only using k-mers then they can compare the accuracies of this model to the NN-RNALoc(noPPI) model to directly estimate the effects of their new distance-based sub-sequence profiles. This would allow the visualization of the increases in accuracy from k-mers, PPI and distance-based features.

6. The utilization of novel features to improve classifier accuracy is very interesting, however it would be equally intriguing to see why these features increase accuracy. For example, what are the most informative distance-based subsequence profiles for each subcellular location? Are some of these, or their respective k-mers enriched for RNA-binding motifs? In addition, are certain subcellular locations enriched for certain protein-protein interactions? I would recommend adding a figure exploring these questions.

Reviewer #2: Authors proposed a deep learning framework for mRNA sub-cellular locations prediction. Authors have proved that information of proteins assists model to predict sub-cellular locations more precisely. The paper seems interesting and will be helpful for biomedical researchers.

6. PLOS authors have the option to publish the peer review history of their article (what does this mean?). If published, this will include your full peer review and any attached files.

Reviewer #1: No

Reviewer #2: No

---

## [Author Response · Author response to Decision Letter 0]

5 Jun 2022

Letter to the associate editor

June 1, 2022

Re: "NN-RNALoc: neural network-based model for prediction of

mRNA sub-cellular localization using distance-based sub-sequence profiles"

Dear Jianhong Zhou,

We would like to bring to your attention that the above manuscript has been revised in

response to the reviewers' comments. In the paper, revisions and new text are highlighted

in blue. Please find enclosed with this letter our response to the reviewers. The revised

manuscript has also been uploaded to the journal's online submission system. We would

like to take this opportunity to thank the referees for their insightful comments and

constructive criticism, which led to substantial revisions to this paper.

Several sections, including the Introduction, Results, and Discussion, were rewritten to

enhance the overall readability of the manuscript. In response to the reviewers'

comments, four figures and one table were added. Each coauthor has approved the final

form of the revision, which was developed in consultation with all coauthors. We

sincerely hope that PLOS ONE will publish the revised manuscript.

Sincerely yours,

Changiz Eslahchi, Rosa Aghdam

Department of Computer Science, Shahid Beheshti University, Tehran, Iran.

School of Biological Science, Institute for Research in Fundamental Sciences (IPM),

Tehran, Iran.

---

## [Decision Letter · Decision Letter 1]

6 Nov 2022

PONE-D-21-31782R1NN-RNALoc: neural network-based model for prediction of mRNA sub-cellular localization using distance-based sub-sequence profilesPLOS ONE

Dear Dr. Eslahchi,

Thank you for submitting your manuscript to PLOS ONE. After careful consideration, we feel that it has merit but does not fully meet PLOS ONE’s publication criteria as it currently stands. Therefore, we invite you to submit a revised version of the manuscript that addresses the points raised during the review process.

We look forward to receiving your revised manuscript.

Kind regards,

Nguyen Quoc Khanh Le

Academic Editor

PLOS ONE

Reviewers' comments:

Reviewer's Responses to Questions

**Comments to the Author**

1. If the authors have adequately addressed your comments raised in a previous round of review and you feel that this manuscript is now acceptable for publication, you may indicate that here to bypass the “Comments to the Author” section, enter your conflict of interest statement in the “Confidential to Editor” section, and submit your "Accept" recommendation.

Reviewer #3: (No Response)

2. Is the manuscript technically sound, and do the data support the conclusions?

Reviewer #3: Partly

3. Has the statistical analysis been performed appropriately and rigorously? 

Reviewer #3: No

4. Have the authors made all data underlying the findings in their manuscript fully available?

Reviewer #3: Yes

5. Is the manuscript presented in an intelligible fashion and written in standard English?

Reviewer #3: Yes

6. Review Comments to the Author

Reviewer #3: In this study, the authors developed an ANN based computational model for localization prediction of mRNA. Following are my major concerns that need to be addressed before acceptance.

1. The authors should cite the existing work on mRNA localization. Following articles must be cited.

Asim, M.N., Ibrahim, M.A., Malik, M.I., Zehe, C., Cloarec, O., Trygg, J., Dengel, A. and Ahmed, S., 2022. EL-RMLocNet: An explainable LSTM network for RNA-associated multi-compartment localization prediction. Computational and Structural Biotechnology Journal.

Meher, P.K., Rai, A. and Rao, A.R., 2021. mLoc-mRNA: predicting multiple sub-cellular localization of mRNAs using random forest algorithm coupled with feature selection via elastic net. BMC bioinformatics, 22(1), pp.1-24.

Wang, D., Zhang, Z., Jiang, Y., Mao, Z., Wang, D., Lin, H. and Xu, D., 2021. DM3Loc: multi-label mRNA subcellular localization prediction and analysis based on multi-head self-attention mechanism. Nucleic Acids Research, 49(8), pp.e46-e46.

Zhang, Z.Y., Yang, Y.H., Ding, H., Wang, D., Chen, W. and Lin, H., 2021. Design powerful predictor for mRNA subcellular location prediction in Homo sapiens. Briefings in Bioinformatics, 22(1), pp.526-535.

Yan, Z., Lécuyer, E. and Blanchette, M., 2019. Prediction of mRNA subcellular localization using deep recurrent neural networks. Bioinformatics, 35(14), pp.i333-i342.

2. The authors compared the accuracy with only two existing tools such as RNATracker an mRNALoc. The other tools (mentioned in comment 1) should also be considered to claim the superiority of the NN-RNALoc.

3. There are several shallow learning (SVM, Random forest, XGBoost, LightGBM etc.) and deep learning models are available. The performance of ANN (used in this study) should be compared with these methods as well.

4. The NN-RNALoc can predict an mRNA to any one localization. However, it is the very fact that a single mRNA could be present in more than one location. So, how the proposed study will address this problem?

5. The area under receiver operating characteristics curve (AU-ROC) and precision-recall curve (AU-PRC) should be included in the performance metrics.

7. PLOS authors have the option to publish the peer review history of their article (what does this mean?). If published, this will include your full peer review and any attached files.

Reviewer #3: No

---

## [Author Response · Author response to Decision Letter 1]

6 Jan 2023

We thank the reviewer for the critical assessment of our work. We

address the concerns point by point in pdf file.

---

## [Decision Letter · Decision Letter 2]

8 May 2023

PONE-D-21-31782R2NN-RNALoc: neural network-based model for prediction of mRNA sub-cellular localization using distance-based sub-sequence profilesPLOS ONE

Dear Dr. Eslahchi,

Thank you for submitting your manuscript to PLOS ONE. After careful consideration, we feel that it has merit but does not fully meet PLOS ONE’s publication criteria as it currently stands. Therefore, we invite you to submit a revised version of the manuscript that addresses the points raised during the review process. Please revise the manuscript according to the comments raised by the reviewers.

We look forward to receiving your revised manuscript.

Kind regards,

Suyan Tian

Academic Editor

PLOS ONE

Journal Requirements:

Reviewers' comments:

Reviewer's Responses to Questions

**Comments to the Author**

1. If the authors have adequately addressed your comments raised in a previous round of review and you feel that this manuscript is now acceptable for publication, you may indicate that here to bypass the “Comments to the Author” section, enter your conflict of interest statement in the “Confidential to Editor” section, and submit your "Accept" recommendation.

Reviewer #4: All comments have been addressed

Reviewer #5: (No Response)

2. Is the manuscript technically sound, and do the data support the conclusions?

Reviewer #4: Yes

Reviewer #5: Yes

3. Has the statistical analysis been performed appropriately and rigorously? 

Reviewer #4: Yes

Reviewer #5: Yes

4. Have the authors made all data underlying the findings in their manuscript fully available?

Reviewer #4: Yes

Reviewer #5: Yes

5. Is the manuscript presented in an intelligible fashion and written in standard English?

Reviewer #4: Yes

Reviewer #5: Yes

6. Review Comments to the Author

Reviewer #4: The localization of messenger RNAs (mRNAs) is a frequently observed phenomenon and a crucial aspect of gene expression regulation. It is also a mechanism for targeting proteins to a specific cellular region. Moreover, prior research and studies have shown the significance of intracellular RNA positioning during embryonic and neural dendrite formation. Incorrect RNA localization, which can be caused by a variety of factors, such as mutations in trans-regulatory elements, has been linked to the development of certain neuromuscular diseases and cancer. In this study, we introduced NN-RNALoc, a neural network-based method for predicting the cellular location of mRNA using novel features extracted from mRNA sequence data and protein interaction patterns. In fact, we developed a distance-based subsequence profile for RNA sequence repres. This work is meaningful in this field. This work can be accepted.

Reviewer #5: 1.Most figures presented in the paper are pixelized. They can be converted to vectorized ones to improve the resolution.

2.The "Evaluation criteria" section should be placed in the Materials and Methods section instead of Results.

3.The tables in the paper are not using standard three-line tables. Please use three-line tables instead.

7. PLOS authors have the option to publish the peer review history of their article (what does this mean?). If published, this will include your full peer review and any attached files.

Reviewer #4: No

Reviewer #5: No

---

## [Author Response · Author response to Decision Letter 2]

10 May 2023

Response to the reviewers

10 May 2023

Manuscript title: NN-RNALoc: neural network-based model for prediction of mRNA sub-cellular localization using distance-based sub-sequence profiles

Manuscript number: PONE-D-21-31782

Revision Version: 3

We thank the reviewer for the critical assessment of our work. In the following, we

address the concerns point by point.

Reviewer #4: The localization of messenger RNAs (mRNAs) is a frequently observed phenomenon and a crucial aspect of gene expression regulation. It is also a mechanism for targeting proteins to a specific cellular region. Moreover, prior research and studies have shown the significance of intracellular RNA positioning during embryonic and neural dendrite formation. Incorrect RNA localization, which can be caused by a variety of factors, such as mutations in trans-regulatory elements, has been linked to the development of certain neuromuscular diseases and cancer. In this study, we introduced NN-RNALoc, a neural network-based method for predicting the cellular location of mRNA using novel features extracted from mRNA sequence data and protein interaction patterns. In fact, we developed a distance-based subsequence profile for RNA sequence repres. This work is meaningful in this field. This work can be accepted.

Thank you for recognizing the significance of our work on NN-RNALoc, a neural network-based method for predicting mRNA localization, and for recommending its acceptance. Thank you for taking the time to review our work, we appreciate your feedback and insights.

Reviewer #5: 1. Most figures presented in the paper are pixelized. They can be converted to vectorized ones to improve the resolution.

Thank you for your feedback. We have taken your suggestion into consideration and have regenerated the figures presented in the paper in vectorized format with high resolution as suggested. We hope that the improved quality of the figures enhances the readability and overall presentation of our work.

2.The "Evaluation criteria" section should be placed in the Materials and Methods section instead of Results.

Thank you for bringing this to our attention. We appreciate your feedback and have made the necessary revisions by moving the "Evaluation criteria" section from the Results section to the end of “Materials and Methods” section.

3. The tables in the paper are not using standard three-line tables. Please use three-line tables instead.

Thank you for your comment regarding the format of the tables presented in our paper. We appreciate your feedback and have revised the table format to adhere to the Plose One template and standard three-line table format.

---

## [Editor Report · Decision Letter 3]

15 May 2023

NN-RNALoc: neural network-based model for prediction of mRNA sub-cellular localization using distance-based sub-sequence profiles

PONE-D-21-31782R3

Dear Dr. Eslahchi,

We’re pleased to inform you that your manuscript has been judged scientifically suitable for publication and will be formally accepted for publication once it meets all outstanding technical requirements.

Kind regards,

Suyan Tian

Academic Editor

PLOS ONE

Additional Editor Comments (optional):

All comments raised by the reviewers have been addressed perfectly, the manuscript is acceptable for being published by the journal.
---

## [Editor Report · Acceptance letter]

19 Jul 2023

PONE-D-21-31782R3 

NN-RNALoc: neural network-based model for prediction of mRNA sub-cellular localization using distance-based sub-sequence profiles  

Dear Dr. Eslahchi:

I'm pleased to inform you that your manuscript has been deemed suitable for publication in PLOS ONE. Congratulations! Your manuscript is now with our production department. 

Kind regards, 

on behalf of

Dr. Suyan Tian 

Academic Editor

PLOS ONE